# Antiretroviral Drug Repositioning for Glioblastoma

**DOI:** 10.3390/cancers16091754

**Published:** 2024-04-30

**Authors:** Sarah R. Rivas, Mynor J. Mendez Valdez, Jay S. Chandar, Jelisah F. Desgraves, Victor M. Lu, Leo Ampie, Eric B. Singh, Deepa Seetharam, Christian K. Ramsoomair, Anna Hudson, Shreya M. Ingle, Vaidya Govindarajan, Tara T. Doucet-O’Hare, Catherine DeMarino, John D. Heiss, Avindra Nath, Ashish H. Shah

**Affiliations:** 1Surgical Neurology Branch, National Institute of Neurological Diseases and Stroke, Bethesda, MD 20892, USA; sxr861@students.jefferson.edu (S.R.R.); la2de@virginia.edu (L.A.); avindra.nath@nih.gov (A.N.); 2Section of Virology and Immunotherapy, Sylvester Comprehensive Cancer Center, University of Miami Miller School of Medicine, Miami, FL 33136, USAebs83@med.miami.edu (E.B.S.);; 3Neuro-Oncology Branch, Center for Cancer Research, National Cancer Institute, Bethesda, MD 20892, USA

**Keywords:** abacavir, antiretroviral, drug repurposing, glioblastoma, lamivudine, reverse transcriptase inhibitors

## Abstract

**Simple Summary:**

The use of antiretroviral therapy has shown promising antineoplastic effects in multiple cancers; however, its efficacy in glioblastoma is unknown. We conducted an unbiased screen of 16 antiretroviral medications in 40 glioma cell lines and validated their efficacy in patient-derived glioma neurospheres and established cell lines. Our study provides the first mechanistic and functional insight into the utility of drug repurposing for malignant gliomas, which supports the current literature. Given their safety profile, preclinical efficacy, and neuropenetrance, antiretroviral therapy may be a promising adjuvant treatment for glioblastoma.

**Abstract:**

Outcomes for glioblastoma (GBM) remain poor despite standard-of-care treatments including surgical resection, radiation, and chemotherapy. Intratumoral heterogeneity contributes to treatment resistance and poor prognosis, thus demanding novel therapeutic approaches. Drug repositioning studies on antiretroviral therapy (ART) have shown promising potent antineoplastic effects in multiple cancers; however, its efficacy in GBM remains unclear. To better understand the pleiotropic anticancer effects of ART on GBM, we conducted a comprehensive drug repurposing analysis of ART in GBM to highlight its utility in translational neuro-oncology. To uncover the anticancer role of ART in GBM, we conducted a comprehensive bioinformatic and in vitro screen of antiretrovirals against glioblastoma. Using the DepMap repository and reversal of gene expression score, we conducted an unbiased screen of 16 antiretrovirals in 40 glioma cell lines to identify promising candidates for GBM drug repositioning. We utilized patient-derived neurospheres and glioma cell lines to assess neurosphere viability, proliferation, and stemness. Our in silico screen revealed that several ART drugs including reverse transcriptase inhibitors (RTIs) and protease inhibitors (PIs) demonstrated marked anti-glioma activity with the capability of reversing the GBM disease signature. RTIs effectively decreased cell viability, GBM stem cell markers, and proliferation. Our study provides mechanistic and functional insight into the utility of ART repurposing for malignant gliomas, which supports the current literature. Given their safety profile, preclinical efficacy, and neuropenetrance, ARTs may be a promising adjuvant treatment for GBM.

## 1. Introduction

Glioblastoma (GBM) is a World Health Organization (WHO) grade IV brain tumor with a poor prognosis of 15 months despite maximal safe resection, radiation, and chemotherapy [1]. Failures in GBM treatment have been partially attributed to its heterogeneous molecular landscape that drives treatment resistance and inevitable recurrence [2]. As a result, there remains a dire need to explore novel treatment regimens that can be readily translated into the clinical setting. Drug repositioning or drug repurposing of existing Food and Drug Administration (FDA)-approved drugs for the treatment of another disease is one such method to explore targeted therapies [3].

Since the advent of antiretrovirals in the 1980s, antiretroviral therapy (ART) has been tested as an adjuvant cancer therapy in numerous malignancies including prostate carcinoma, non-small-cell lung cancer, glioma, and breast cancer [4,5,6]. Investigation of ART in cancer began due to its ability to reduce morbidity and mortality in Acquired Immunodeficiency Syndrome (AIDS)-related malignancy, Kaposi’s sarcoma, independent of Human Immunodeficiency Virus (HIV) viral load [5]. Additionally, the pleiotropic effects of ART have been reported in other cancers where it has been found to inhibit angiogenesis, cell invasion, and proliferation [7,8,9,10]. Recent studies have also demonstrated that combinatorial regimens of ART sensitize tumors to adjuvant chemotherapy and radiation [11,12]. However, the vast majority of these studies have been conducted on systemic solid tumors and/or HIV-associated malignancies, and only rarely has the use of ART been reported for gliomas.

To better understand the drug repositioning potential of ART in GBM, we used a novel in silico approach to conduct an unbiased neuropharmacological screen and assessed both transcriptomic and phenotypic changes in vitro with both patient-derived and established GBM cell lines with candidate ART. Finally, we conducted a systematic review of the current literature to contextualize our results in the ultimate setting of translating ART in contemporaneous neuro-oncology.

## 2. Materials and Methods

### 2.1. DepMap

To identify potential ART drug repositioning candidates, we queried the Broad Institute’s Cancer DepMap portal to compare drug sensitivity for glioma cancer cell lines to selected ART drugs. The primary DepMap screen (PRISM) applies a fixed drug concentration (2.5 µM) of the target drug and calculates a logFC drug response ratio relative to DMSO controls. LogFC values less than −1 imply cell-line-specific drug sensitivity at the selected dose. Pearson correlations for cell lines and antiretroviral medications were conducted with the DepMap Data Explorer (https://depmap.org/portal/interactive/). Each cell line was plotted against each antiretroviral medication, and the Pearson correlations were extrapolated for analyses. Likewise, all the antiretroviral medications explored in this study were plotted against each other, and their respective Pearson correlations were extrapolated for further analysis. 

### 2.2. sRGES Score and Disease Signature

Summarized reversal of gene expression scores (sRGESs) for glioblastoma accurately predict drug efficacy based on the reversal of transcriptional cellular signatures [13]. GBM signatures from The Cancer Genome Atlas (TCGA) were matched to samples from the Cancer Cell Line Encyclopedia (CCLE) as described previously [14]. The profiles were compared to preprocessed RNA-seq data from human frontal lobe tissue from the Genome Tissue Expression Project (GTEX) to determine differential expression. Drug expression profiles were characterized by accessing data from the LINCS L1000 assay to determine compound-induced changes in gene expression, as shown previously by Shah et al. [6]. Gene expression counts and Central Nervous System Multiparametric Optimization (CNS-MPO) values were also attained by filtering the data for antiretroviral therapies. Data were visualized using Microsoft Excel v16.84 and the pheatmap function in R v4.2.3.

### 2.3. Cell Culture

Non-adherent patient-derived primary GBM neurospheres (GBM28, GBM43) were acquired from both the Mayo Clinic Patient-Derived Cell line Repository, and adherent culture cells (A172 and U87) were obtained from American Type Tissue Culture Collection (ATCC, Manassas, VA, USA) [15]. GBM neurospheres were cultured in serum-free media in Dulbecco’s modified Eagle’s medium (DMEM/F-12) without phenol red, supplemented with 2% B27 (Invitrogen, Waltham, MA, USA), 1% penicillin/streptomycin, and 20 ng/mL of basic fibroblast growth factor and human epidermal growth factor (50 µL). A172 and U87 cell lines were cultured in DMEM with 5% FBS supplemented with 1% penicillin/streptomycin. Neurospheres and adherent culture cells were incubated at 37 °C (5% CO_2_) and neurospheres were maintained in low-attachment flasks. Both adherent established cell lines and patient-derived neurospheres were dissociated using Trypsin EDTA 0.25% and TrypLE dissociation solution respectively. 

### 2.4. Cell Viability and Proliferation Assay

Cells were seeded at 5000 cells/well (100 cells/µL) in a 96-well plate. A172, GBM28, and GBM43 were treated with varying doses of abacavir (ABC), lamivudine (LMV), raltegravir (RLT), indinavir (IND), or darunavir (DAR) and kept in 37 °C for 4 days. ART drugs were obtained from the NIAID HIV Reagent Program (Manassas, VA, USA). Viability was measured with an XTT cell proliferation assay kit according to the manufacturer’s instructions (ATCC, VA, USA). For the proliferation assay, cells were seeded at 1000 cells/well in a 96-well E-plate. A172 and U87 cell lines were treated with 20 μM abacavir (ABC) or 20 μM lamivudine (LMV) and kept at 37 °C for 48 h. This was determined based on previous work our group conducted using a dose–response curve to abacavir [16]. Cell proliferation was measured with the xCELLigence RTCA DP instrument according to the manufacturer’s instructions (ACEA Bioscience, San Diego, CA, USA).

### 2.5. Immunofluorescence

Cells were cultured as indicated above and treated with ABC and LMV for 48 h. Cells were seeded in 2-chambered slides at a concentration of 2.5 × 106 cells/cm^2^. Neurospheres adhered to the slide using serum-free Geltrex (ThermoFisher, Waltham, MA, USA). Cells were fixed with 4% paraformaldehyde (ThermoFisher) for 20 min and washed with phosphate-buffered saline (PBS). To permeabilize the cell membrane, cells were incubated with 0.01% Triton X for 5 min, washed with PBS, and blocked with 10% normal goat serum and 2% bovine serum albumin (BSA) (BioRad, Hercules, CA, USA). Primary antibodies were incubated overnight and washed with PBS. Finally, slides were incubated in fluorescent-tagged goat anti-rabbit and goat anti-mouse antibodies with 2% normal goat serum (NGS) and 2% BSA using the following antibodies: Plectin Rab IgG (Abcam, Boston, MA, USA), vimentin Mab IgG1 (Aligent DAKO, Santa Clara, CA, USA), OCT-4 (Millipore Sigma, Burlington, MA, USA), Goat anti-rabbit, and Goat anti-mouse (ThermoFisher). Foci quantification was calculated using ImageJ v1.53 in biological and technical triplicate and quantified using one-way ANOVA (*p* < 0.05). 

### 2.6. Systematic Review 

A systematic online literature review was conducted in October of 2022 using 1 electronic database (PubMed). The review was conducted in compliance with the Preferred Reporting Items for Systematic Reviews and Meta-Analyses (PRISMA) guidelines and recommendations (20171303). The search terms, “clinical trials”, “HAART”, “highly active antiretroviral therapy”, and “cancer” were combined to include the initial cohort of papers for Table 1. To include studies in this review, the following search terms were used to construct Table 2, “glioma”, “In Vitro”, “In Vivo”, “Abacavir”, “Emtricitabine”, “Lamivudine”, “Tenofovir”, “Zidovudine”, “Doravirine”, “Efavirenz”, “Nevirapine”, “Rilpivirine”, “Atazanavir” “Darunavir”, “Fosamprenavir”, “Ritonavir”, “Enfuvirtide”, “Maraviroc”, “Cabotegravir”, “Dolutegravir”, “Raltegravir”, and “Fostemsavir”. We then reviewed the reference lists of all retrieved articles for identification of potentially relevant cases. The following selection criteria were used by independent investigators to assess the remaining articles. Eligible cases to be included in Table 1 were required to meet the following criteria: (1) a cancer patient population, (2) ART was used to try and treat the cancer directly, and (3) reported survival outcome data. Papers that described patients who were on ART prior to cancer diagnosis for HIV were excluded. Eligible cases to be included in Table 2 were required to meet the following criteria: (1) glioma-based cell line or animal model use was essential and (2) ART was used to treat cell lines or animal models. For all studies, the full article needed to be available and reported in English.

## 3. Results

### 3.1. Antiretroviral Therapy Alters the Transcriptomic Landscape in GBM

To assess the efficacy of ART on GBM cell viability, we conducted a pharmacological screen using the Cancer Dependency Map Portal (DepMap) portal of 16 antiretroviral drugs in established glioma cell lines. Of the 16 antiretrovirals screened, abacavir and ritonavir demonstrated consistent inhibition of cell viability on the 40 glioma cell lines with the highest efficacy in U87 and SNU201 cell lines (Figure 1A). Additionally, we employed a previously validated tool that recapitulates perturbagen-associated gene effects in target cell populations: the summarized Reversal of Gene Expression Score (sRGES) [13]. A negative sRGES strongly correlates to reduced cell viability across multiple tumor types including GBM [14]. Using this pipeline, we filtered 11,951 drugs from the LINCS1000 dataset for antiretroviral compounds and identified 4 ART compounds. The sRGES values from antiretroviral data demonstrated that all antiretrovirals demonstrated a negative sRGES for GBM, indicating a reversal of the gene signature of glioblastoma aggregate samples—lamivudine (−0.0235), efavirenz (−0.0679), tenofovir (−0.123), and ritonavir (−0.630) (Figure 1B). These highly reversed genes were especially downregulated by maraviroc, which decreased the expression of CHEK2 and CDCA4 which code for proteins involved in DNA damage repair and cell proliferation (Figure 1C). Lastly, we further stratified these compounds using the CNS-MPO index to determine the optimal pharmacokinetic profile and blood–brain barrier penetrance. A CNS-MPO score > 4 has been used to determine if target compounds reside in an optimal neuropharmacological niche [14]. Of these compounds, efavirenz and ritonavir demonstrate a CNS-MPO value greater than 4, suggesting a role for translating these compounds in neuro-oncology (Figure 1D).

### 3.2. ART Decreases GBM Cell Viability

To validate these effects of ART on GBM in vitro, we assessed changes in cell viability on GBM neurospheres and established glioma cell lines. Cells were treated with six antiretrovirals, including abacavir (ABC), lamivudine (LMV), raltegravir (RLT), indinavir (IND), or darunavir (DAR) for 4 days. ABC and LMV indicated a decrease in viability in both neurospheres and A172 cells, respectively (Figure 2A). Violin plots of the antiretroviral effect on various cancer cell lines showed the greatest decrease in viability with ABC, LMV, and RLT. (Figure 2B) In particular, the NRTI, ABC, demonstrated the largest negative log 2-fold change in GBM cell lines. (Figure 2B).

### 3.3. Antiretroviral Therapy Decreases Stemness In Vitro

Glioblastoma intratumoral heterogeneity and outcomes are directly correlated to the GBM stem cell burden. Previously, it has been demonstrated that antiretroviral drugs reduce cellular stem cell markers [31,32]. Therefore, we sought to investigate if our compounds affected markers of tumor stemness. A172 and U87 cell lines were treated with ABC 20 μM or LMV 20 μM for 48 h and stained for OCT4 and vimentin using immunofluorescence. Both ABC and LMV induced a marked decrease in OCT4 and vimentin in both GBM cell lines compared to naïve and sham-treated controls (Figure 3A,B). Quantification revealed a significant depletion of OCT-4 and vimentin foci across all treatments compared to controls (Figure 3C,D). Cell index (CI) values obtained from the xCELLigence RTCA demonstrated that LMV and ABC had an antiproliferative effect on A172 and U87 cells, which was greatest after ABC (20 μM) (Figure 3E,F).

### 3.4. Multimodal ART Treatment May Be Synergistic against GBM

Since many antiretroviral drugs are given in combination, we subsequently screened therapeutic combinations of integrase, protease, reverse transcriptase, and entry inhibitors. The combination therapy of darunavir–atazanavir and nevirapine–emtricitabine demonstrated the strongest synergistic antineoplastic effects when compared to other ART combination therapies. (Figure 4A). Previously, ARTs have shown the ability to mitigate cellular proliferation, stemness, migration, and invasion. Figure 4B illustrates RTI involvement in decreasing tumor proliferation and promoting differentiation. Additionally, PIs decrease the capability to migrate and invade by decreasing matrix metalloproteases and VEGF expression in gliomas. To understand how ART alters transcriptomic programs in GBM, we assessed ARTs and drugs from NCI-supported GBM clinical trials for synergy with temozolomide (TMZ) and transcription reversal utilizing data from SynergySeq [33]. ARTs demonstrated similar disease discordance to drugs such as bromodomain inhibitors (I-BET-151, I-BET-762) and EGFR inhibitors (erlotinib and gefitinib). Antiretroviral agents such as lopinavir demonstrated greater discordance than others, while some agents showed greater similarity in the transcriptional response to TMZ (Figure 4C). 

### 3.5. Systematic Review

To better understand the previous work regarding drug repositioning of antiretrovirals in cancer, we conducted a systematic review of the literature to evaluate both clinical and preclinical studies involving both gliomas specifically and then the broader cancer demographic. Concerning glioma and general cancer studies, we identified 72 and 1303 studies for evaluation, respectively (Figure 5). After final screening for inclusion/exclusion criteria, a total of 7 clinical and 10 preclinical studies met the selection criteria and were analyzed. The characteristics of the included articles and their outcomes are summarized in Table 1 and Table 2. 

Among the three clinical trials, two showed response rates were reported by the studies and ranged from 11% to 65%. The most common adverse effects seen were GI upset and skin irritation. From these studies, overall survival ranged from 40 weeks to 12 months, and progression-free survival ranged from 5.5 to 6 months (Table 1). Among the 10 preclinical studies, the most common drug investigated was ritonavir. Overall, ART decreased tumor cell viability, induced apoptosis, and reduced tumor progression in in vivo models. Of these studies, five reported in vivo models with an increased overall survival and reduced tumor progression following ART treatment. 

## 4. Discussion

The use of antiretrovirals in oncology has been investigated as combinatorial therapies and monotherapies in several cancers including multiple myeloma, acute myelogenous leukemia, adenoid cystic carcinoma, and bladder cancer [17,19,21,34,35,36,37]. We conducted this hybrid study to improve our understanding of molecular and phenotypic underpinnings of ART for GBM since comprehensive drug repositioning studies have been limited. Outcomes for GBM remain poor due to the molecular and cellular heterogeneity that is driven by undifferentiated GBM stem cell populations. Since the GBM stem cell niche dictates treatment resistance and tumor recurrence, ART therapy remains a promising adjuvant therapy for GBM due to its pleiotropic effects on cancer stemness and proliferation [38]. 

Ritonavir (RTV) and other protease inhibitors exhibited potent antitumor activity for GBM and have also been studied in early-stage clinical trials [12]. Ahluwalia et al. performed a phase II clinical trial that assessed combinatorial ritonavir/lopinavir in gliomas. Despite previous research that supported these findings in vitro, only 11% of patients experienced a 6-month progression-free survival (PFS), which is not significantly different from those on standard-of-care treatment. This lack of clinical response may be attributed to subtherapeutic CNS dosing, a lack of combinatorial ART drugs, and candidate drug selection [21,22]. Hoover et al. performed a phase II clinical trial that treated patients diagnosed with recurrent adenoid cystic carcinoma (ACC) with nelfinavir. Overall, they showed a modest prolongation of the progression-free survival rate of 5.5 months compared to the standard 3.5. However, the authors concluded that nelfinavir was not sufficient as a monotherapy and would be better assessed in combination with standard-of-care treatment [19].

Recent investigations of PIs in cancer and specifically their use in GBM have focused on combinatorial regimens. Rauschenbach et al. showed that isolated RTV exhibited cytostatic and anti-migratory effects, while in combination with TMZ, it worked synergistically in glioblastoma [12]. Similarly, Azzalin et al. showed synergy between RTV and TMZ, and RTV decreased the carmustine treatment dose five-fold in vivo [25]. Other PIs, such as nelfinavir, have also been studied in early Phase I trials for GBM with a demonstrable safety profile. Overall survival in this cohort was similar to standard-of-care regimens, albeit most of the tumor recurrence occurred outside of the treatment field. In our preclinical studies, PIs such as darunavir and indinavir demonstrated moderate efficacy in decreasing GBM neurosphere viability but were outperformed by reverse transcriptase inhibitors. 

Previously, reverse transcriptase had been proposed as a target for cancer therapy since undifferentiated cancer cells tend to express higher levels of RT than differentiated cells [39,40]. In other cancer cell lines, inhibition of endogenous RT with efavirenz and nevirapine reduced proliferation and exhibited cytostatic effects, suggesting a role of RTI drug repurposing for oncology [41]. Our analysis of 16 ARTs in 40 glioma cell lines revealed several candidates for in vitro experiments, including the RTIs ABC and LMV. Using pharmacogenetic screening based on LINCS1000 signature profiles, we also identified that most ART drug regimens reverse the GBM disease signature, which is a hallmark of drug efficacy. sRGESs summarize the ability of a drug to reverse the cancer-specific transcriptomic signature; this tool has been previously validated in several cancers, including GBM, and correlates strongly with in vitro cytotoxicity. For GBM specifically, NRTIs (ABC and LMV) reduced cell viability and decreased stemness markers in vitro in GBM cell lines. 

RTIs have also been demonstrated to reduce expression of the stem cell marker, Human Endogenous Retrovirus-K. Recently, human endogenous retroviruses (HERVs) have been implicated in the promotion of stem cell phenotypes in a variety of cancers and neurological diseases [42]. Although epigenetically repressed in normal tissues, the HERV-K (HML-2) env protein is specifically overexpressed in many tumor types, and correlates to a cancer stem cell phenotype [43,44,45,46]. Additionally, Berkhout et al. first identified the active reverse transcriptase (RT) enzyme by human endogenous retrovirus-K (HML-2) [47].

Our group previously demonstrated the ability of abacavir to decrease the activity of HERV-K (HML-2) RT as well as the stem cell marker OCT4 [16]. Given the preliminary data demonstrating the capacity of RTIs to decrease markers of stemness and diminish the activity of RT, more thorough and widespread investigations with an expanded antiretroviral medication regimen are merited. Our in silico analysis using the DepMap portal was limited to the antiretroviral medications available in that database, and therefore, did not include all currently available ARTs. 

RTIs have demonstrated marked efficacy in vitro and in vivo in CNS tumors such as medulloblastoma [48]. It was considered that RTIs could secondarily inhibit DNA replication and reduce telomerase activity [49]. Telomerase reverse transcriptase (TERT) mutations are prevalent in high-grade gliomas and can be specifically targeted with NRTIs through RNA-dependent RNA polymerase blockade [50]. TERT activity has been implicated in promoting stem-like features in cancer and its inhibition by RTI promotes cell differentiation. GBM cell line A172 exhibited chromosome instability and cytotoxicity post TERT inhibition [49,50], giving additional evidence that further investigations into NRTIs are merited. Future studies assessing the efficacy of ABC and LMV in TERT inhibition could strengthen the evidence for ART use and provide additional avenues through which these medications may reduce stemness in GBM.

## 5. Limitations

A limitation in the clinical translation of ART for the treatment of GBM is penetration of the blood–brain barrier (BBB). To overcome this limitation, we incorporated an analysis of the CNS-MPO score, which is a validated tool to identify drugs that reside in an optimal neuropharmacological niche with acceptable blood–brain barrier penetrance. The CNS-MPO score > 4 suggests that these drugs can be candidates for drug repositioning for neurological diseases. Our data indicated a mix of CNS penetration with the highest CNS-MPO scores held by RTV and EFV (CNS-MPO score: RTV = 5.83; EFV = 4.12). Abacavir and lamivudine did not reach a CNS-MPO score > 4; however, most ARTs can penetrate the CNS at low concentrations. Clinical translation of ART may be readily feasible with combination therapies, intraventricular ART administration, intratumoral therapy, or by implementing surgical strategies to open the BBB. However, long-term continuous therapy is likely required to facilitate long-term disease control. Similar to HIV, ART therapy for GBM may require a multiple-drug regimen to overcome tumor persistence and drug resistance pathways. The triple drug therapy of one protease inhibitor and two RTIs has been shown to longitudinally suppress HIV replication and eliminate cellular reservoirs. Similarly, combinatorial ART approaches are likely required in addition to chemoradiation to afford the best chance at disease control.

## 6. Conclusions

In the context of the current literature, our findings support the emerging evidence of ART as an adjuvant treatment for GBM. Although there was no universal summary statistic, the in vitro data published to date demonstrate the ability of ART to decrease tumor viability, proliferation, and stemness. With respect to preclinical studies of other tumor models, the consistent prolongation of overall survival indicates the translational potential of ART repositioning to treat GBM in the future.

## Figures and Tables

**Figure 1 cancers-16-01754-f001:**
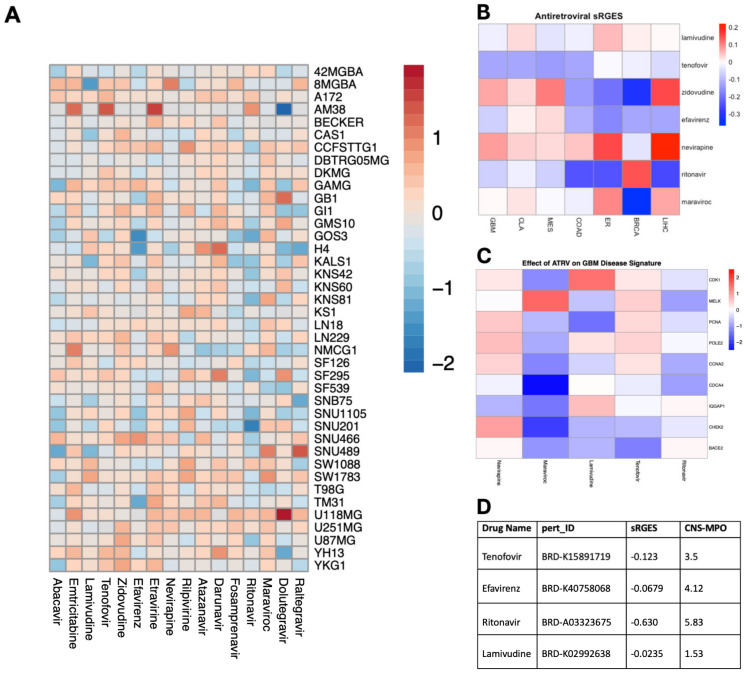
Unbiased screen of ART identifies candidate drugs with potent anti-glioma activity. (**A**) Heatmap illustrating the impact of antiretroviral drugs on the survival of individual glioma cell lines in terms of log_2_fold change (range = −2.13, 1.92, SD= 0.4692). Data sourced from the PRISM Repurposing Screen and analyzed using DepMap, with warmer colors indicating improved survival and cooler colors indicating reduced survival. (**B**) Heatmap depicting the capacity of individual antiretroviral drugs to reverse the genetic signature of various primary cancers based on sRGESs calculated using drug expression profiles accessed from the LINCS L1000 assay (range = −0.37, 0.22, SD = 0.124). (**C**) Heatmap illustrating the impact of select antiretroviral drugs on the expression of 9 genes critically associated with reversal of genetic signature in glioblastoma. (**D**) Table supplying sRGES and CNS-MPO values for each antiretroviral drug to inform both the effect of each drug on genetic signature reversal and the drug’s optimal pharmacokinetic profile [sRGES range: −0.0235, −0.630, SD: 0.282], [CNS-MPO range: 1.53, 5.83, SD: 1.78].

**Figure 2 cancers-16-01754-f002:**
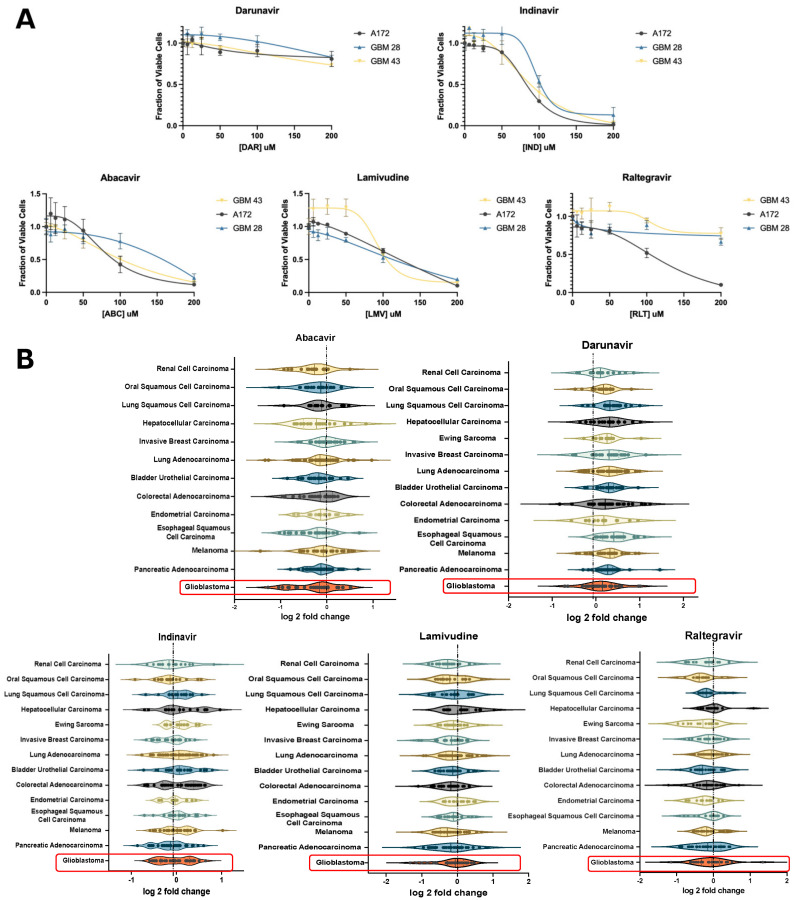
In vitro validation of candidate ARTs. (**A**) XTT assay of patient-derived glioblastoma cell lines (GBM 28, 43) and a pure glioma cell line (A172) showing the effect of select antiretrovirals on cell viability relative to increased dosing. (**B**) Violin plots depicting the impact of select antiretroviral drugs on the survival of established cell lines stratified based on primary cancers in the Cancer Dependency Map database with greater than 10 established cell lines. This demonstrates the relative effect of each antiretroviral drug on the log_2_fold change in survival for each primary cancer. Highlighted plots represent GBM and demonstrate a median log_2_fold change of less than 0 when treated with abacavir, lamivudine, and raltegravir.

**Figure 3 cancers-16-01754-f003:**
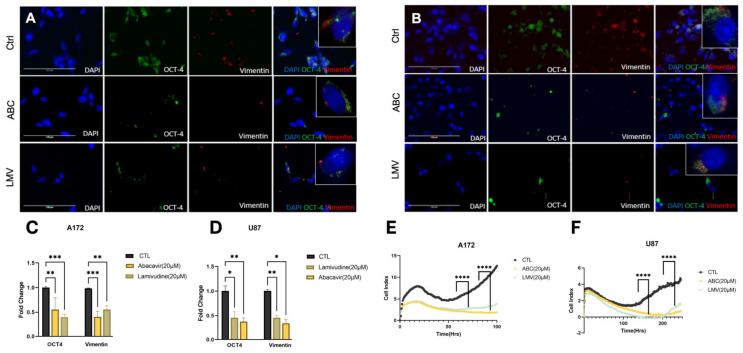
Reverse transcriptase inhibitors decrease stemness and self-renewal capacity. (**A**,**B**) Identified foci of stemness markers using IF of A172 and U87 cells treated with ABC 20 μM and LMV 20 μM for 48 h. OCT-4 (green), vimentin (red), and DAPI. (**C**,**D**) Quantification of fold change of foci shows a significant decrease in expression after treatment. (**E**,**F**) Cell proliferation assessed by xCelligence assay of A172 and U87 cell lines while treated with ABC 20 μM and LMV 20 μM: RTIs significantly decreased proliferation compared to control. (ANOVA, r * *p* < 0.05, ** *p* < 0.01, *** *p* < 0.001, **** *p* < 0.0001).

**Figure 4 cancers-16-01754-f004:**
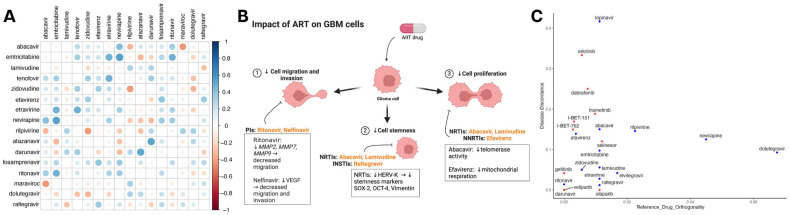
Synergistic effects of ART against the genetic landscape in GBM. (**A**) Heatmap depicting a correlation matrix that represents the comprehensive evaluation of antiretroviral drug efficacy against glioma cell lines using the Cancer Dependency Map (DepMap) database. In total, 16 antiretroviral drugs were identified from 4518 drugs in the PRISM Repurposing Screen. Each cell indicates the strengths of the Pearson correlation coefficient (r) between two antiretrovirals, highlighting their comparative effectiveness. More positive values signify a strong positive correlation, suggesting similar effectiveness, while more negative values indicate more distinct efficacy profiles of the compared drugs. (**B**) Schematic of ART capacity to decrease proliferation, stemness, migration, and invasion in CNS tumors. Figure depicts RTI involvement in decreased proliferation and promoting differentiation. Additionally, PIs are involved in mitigating migration and invasion. (**C**) Data obtained from SynergySeq used to evaluate drugs from the LINCS 1000 small molecules database for disease transcriptional response signatures. Scatterplot of all ARTs (blue) and drugs from recent and current NCI-supported GBM clinical trials (red) that were identified in the database depicting similarity to the transcriptional response of GBM to TMZ (*x*-axis) and degree of glioblastoma (GBM) transcriptional signature reversal (*y*-axis) derived from an independent cohort of 71 patients and matched normal brain tissue.

**Figure 5 cancers-16-01754-f005:**
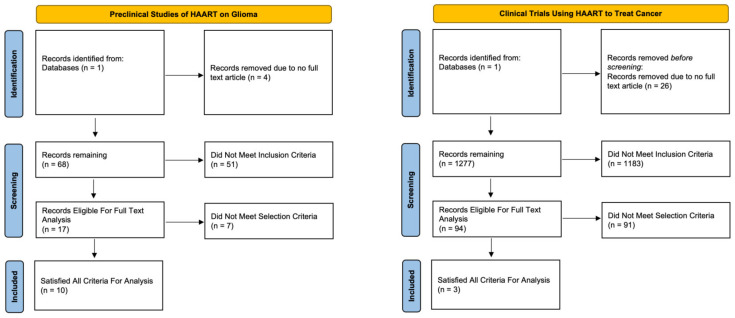
Systematic review flow chart for preclinical trials utilizing antiretroviral medications for the treatment of established glioma cell lines and clinical trials utilizing antiretroviral medications for the treatment of various cancers.

**Table 1 cancers-16-01754-t001:** Clinical trials using HAART to treat cancer.

Author/Year	# of pts	Type of Cancer	HAART Used	Overall Survival (OS)	Response Rate	Adverse Effects	Median OS
Driessen, 2018 [17]	34	Multiple Myeloma	Nelfinavir	12 months	65%	Anemia	5.2 years [18]
Hoover, 2015 [19]	15	Adenoid Cystic Carcinoma	Nelfinavir	Progression-free furvival 5.5 months	Seven patients with stable disease	Hyponatremia Thrombocytopenia Dizziness	17.7 years [20]
Ahluwalia, 2011 [21]	19	Progressive or Recurrent High-Grade Glioma	Ritonavir/ Lopinavir	Progression-free furvival 6 months	11%	Diarrhea Hypercholesterolemia Fatigue	Grade III—10 months Grade IV—6 months [22]

**Table 2 cancers-16-01754-t002:** Preclinical studies of HAART on glioma.

Author/Year	Study Type	Cell Line	Antiretroviral	Study Goals	Conclusions
Novak, 2020 [23]	in vitro	U373	Maraviroc	Examining the role of CCL5 and CCR5 in the tumor microenvironment	CCL5/CCR5 axis could be targeted by maraviroc
Rauschenbach, 2020 [12]	in vivo	BN023; BN197; GNV019; AHNP155; AHNP167; AHNP189	Ritonavir	Combination therapy of antiretrovirals and temozolomide	Increased overall survival in xenograft models
Basile, 2018 [24]	in vitro	LN229; U251	Lopinavir and lopinavir–NO	Examining the effect of lopinavir vs. lopinavir–NO on tumor viability	Lopinavir–NO reduced tumor viability
Azzalin, 2017 [25]	in vitro and in vivo	U87MG; Hu197; GBM-P1	Indinavir and ritonavir	Combination therapy of antiretrovirals and BCNU and temozolomide	Ritonavir and BCNU increased overall survival
Laudati, 2017 [26]	in vitro	C6	Maraviroc	Effect of CCR5 receptor blockade via maraviroc on microglia polarization	CCR5 blockade reduced microglia migration
Kast, 2016 [27]	in vitro	GAMG	Ritonavir; Aprepitant; Temozolomide	Combination therapy of antiretroviral and antiemetic drugs with temozolomide	Ritonavir, aprepitant, and temozolomide exert antitumor action
Funes, 2015 [28]	in vitro	U251MG; SH-SY5Y	Efavirenz	Effect of efavirenz on mitochondrial respiratory function	Inhibition of mitochondrial respiration
Khan, 2010 [29]	in vitro and in vivo	U87MG; NGC407	Azidothymidine (AZT)	Combination therapy of thymidine kinase 1 (toTK1) and AZT	Increased overall survival
Pore, 2006 [30]	in vivo	U87MG; U251MG	Nelfinavir and Amprenavir	Antiretroviral therapy on VEGF and HIF-1α expression and on angiogenesis	Decreased VEGF and HIF-1α expression as well as angiogenesis

## Data Availability

Available upon reasonable request.

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
