# Peer review of "Antiretroviral Drug Repositioning for Glioblastoma"

_cancers, 2024, doi:10.3390/cancers16091754_

Round 1

Reviewer 1 Report

Comments and Suggestions for Authors

The research article “Antiretroviral Drug Repositioning for Glioblastoma” presents result of the computational analysis that identified antiretroviral drugs (ART drugs) as potential drugs for glioblastoma (GB). Next, the authors validated via viability assay several candidate ART drugs, abacavir (ABC), lamivudine (LMV), raltegravir (RLT), indinavir (IND), and darunavir (DAR) in vitro on glioblastoma adherent cell lines as well as neurospheres. They also assessed impact of ART drugs on marker of “stemness”, OCT4, in GB cells. Next, the authors assessed therapeutic combinations of ART drugs and others. Finally, they performed systematic literature review on the subject of study, with particular focus on clinical trials, and discussed the issue of BBB permeability for ART drugs.

Please find below my comments and suggestions.

In case of computational analysis, is it feasible to describe (or provide a link to the code repository, if applicable) all key steps of analysis, if not provided in details in the publications quoted?

If ART drugs target cancer stem cells (CSCs), viability assay is not the best assay to validate their action, as phenotype of GB cells might shift from “stem cell-like” towards the “less stem cell-like” without affecting cell viability. Furthermore, the more clinically efficient drugs might be the ones significantly affecting CSCs without affecting GB cell viability (not necessarily, but possibly).

Only ABC and LMV were selected for ICH analysis of “stemness” markers following treatment. Also, only a few markers were chosen (OCT4 and Vimentin).

Line 154. “Antiretroviral therapy alters the genetic landscape in GBM”.

Definitely, such therapy alters TRANSCRIPTOME (gene expression) landscape, not genetic landscape (unless it alters patterns of incorporation of retrobiome in the DNA of GB cells, which is a possibility, but was not assessed in this study).

In case of ART drug validation, what did you use as a “positive control” to demonstrate that at given concentration ART compound indeed inhibits RT?

Figure 4.B. The font is too small and almost illegible, could you please increase it?

Overall, the article is of definite interest and should be published.

Author Response

In case of computational analysis, is it feasible to describe (or provide a link to the code repository, if applicable) all key steps of analysis, if not provided in details in the publications quoted?

Thank you for your comment. The link to the repository where the computational analysis was done can be found here: https://depmap.org/portal/

The data explorer tool was used to screen for the compounds, and the cell line selector tool was used to screen for the cell lines. We selected the desired compounds and cell lines, and then plotted them against each other using the software and analysis tools found in the depmap portal.

If ART drugs target cancer stem cells (CSCs), viability assay is not the best assay to validate their action, as phenotype of GB cells might shift from “stem cell-like” towards the “less stem cell-like” without affecting cell viability. Furthermore, the more clinically efficient drugs might be the ones significantly affecting CSCs without affecting GB cell viability (not necessarily, but possibly).

Thank you for your comment. In our manuscript, we first highlighted the effects of ART on cell viability outside of the potential of GBM cells shifting to less stem cell-like states. Later on in our work, we showed specifically the potential of ART to decrease the expression of certain markers of stemness, Vimentin and OCT4, thereby indicating that the medication has an impact on shifting the GBM cells towards a less stem cell-like state.

Only ABC and LMV were selected for ICH analysis of “stemness” markers following treatment. Also, only a few markers were chosen (OCT4 and Vimentin).

Thank you for your comment. While it is true that these were the only drugs and markers of stemness chosen, this manuscript was not meant to depict an exhaustive analysis of all markers of stemness and all antiretroviral medications. This project aimed to provide the first line of evidence that antiretroviral medications can reduce the expression of markers of stemness, and merit further exploration.

Line 154. “Antiretroviral therapy alters the genetic landscape in GBM”.

Definitely, such therapy alters TRANSCRIPTOME (gene expression) landscape, not genetic landscape (unless it alters patterns of incorporation of retrobiome in the DNA of GB cells, which is a possibility, but was not assessed in this study).

Thank you for your comment. The wording has been changed in the manuscript.

In case of ART drug validation, what did you use as a “positive control” to demonstrate that at given concentration ART compound indeed inhibits RT?

Thank you for your comment. Reverse transcriptase inhibitors have been widely tested and proven to inhibit RT in the literature. Thus, we did not test their mechanism of action with a positive control.

Figure 4.B. The font is too small and almost illegible, could you please increase it?

Overall, the article is of definite interest and should be published.

Thank you for your comment. The size of the font has been increased.

Reviewer 2 Report

Comments and Suggestions for Authors

The authors ran an unbiased screen of antiretroviral drugs to find promising drug candidates for GBM treatment, and provided some evidence why antiretroviral drugs may be efficacious against GBM, i.e. reduced proliferation and GBM stem cell markers. It makes sense to evaluate the potential of these drugs to improve GBM treatment given the dismal outcome of current therapies for GBM, the literature indicating that antiretroviral drugs could be repositioned for cancer treatment as well as the known safety profile of the antiretroviral drugs. The authors found promising agents (mainly through in silico approaches) and correctly pointed out the limitations of potential ART therapy as adjuvant treatment for GBM. Ultimately, to find more evidence that the suggested drugs/ drug combinations could be efficacious against GMB, the drugs should be extensively tested/in vitro/ and importantly, in orthotopic mouse models of GBM .

It is hard to read the text in the figures, particularly Fig.2B,  Fig.4A,C.

Author Response

The authors ran an unbiased screen of antiretroviral drugs to find promising drug candidates for GBM treatment, and provided some evidence why antiretroviral drugs may be efficacious against GBM, i.e. reduced proliferation and GBM stem cell markers. It makes sense to evaluate the potential of these drugs to improve GBM treatment given the dismal outcome of current therapies for GBM, the literature indicating that antiretroviral drugs could be repositioned for cancer treatment as well as the known safety profile of the antiretroviral drugs. The authors found promising agents (mainly through in silico approaches) and correctly pointed out the limitations of potential ART therapy as adjuvant treatment for GBM. Ultimately, to find more evidence that the suggested drugs/ drug combinations could be efficacious against GMB, the drugs should be extensively tested/in vitro/ and importantly, in orthotopic mouse models of GBM .

Thank you for your comment. The aim of this manuscript was to provide the introductory justification for the use of ART in GBM, with the hope that it will help expand research in the field to extensively research these compounds.

It is hard to read the text in the figures, particularly Fig.2B,  Fig.4A,C.

Thank you for your comment. The font size in the figures has been increased.

Reviewer 3 Report

Comments and Suggestions for Authors

Author Response

Despite the importance and innovation of the topic, I am unfortunately sorry to say that I have to reject the article. The reason is that the work is disjointed, appearing as three separate studies forced together. The first part, although very interesting in itself, is not actually validated in cellular lines because the drugs themselves are not validated. The functional second part involves the use of antivirals on glioblastoma cells, with the rationale being to validate the predictions made in the first part when the authors perform drug repositioning. However the same drugs are not then validated, except for one. Why is this? The third part of the article seems more like a review than a proper conclusion.

Thank you for your comment. For the first part, the screening of the candidate drugs on glioma cells lines is sourced from the DepMap portal, which is data that has been validated through a wide variety of experimental methods. More information can be found here: https://depmap.org/portal/home/#/our-approach

For the second part, we validated two drugs, abacavir and lamivudine. This was based on the initial in silico screening, as well as the availability of the drug for in vitro testing. It was not feasible to validate all of the drugs screened due to time and financial constraints.

Finally, the third portion was meant as a systematic review to highlight the lack of clinical validation of these drugs as adjuvant therapy for GBM. Given the lack of progress in the field, novel approaches are merited, and this is an approach that has not been studied sufficiently.

Comments:

  • Line 115: Why was the dose of 20uM chosen? Was a dose-response curve performed? If so, it should be described.

This was determined based on previous work our lab has published on a dose response curve to abacavir. This has been added to the manuscript.

  • . Line 130: How exactly was this done to assume that the foci are diminishing and not the number of cells?

The quantification can be interpreted as diminishing foci since each trial was done with a known number of cells (2.5 x 106 cells/cm2) that were fixed in 4%PFA and then blocked. The steps of this are described in the manuscript, lines 124-137.

  • . Line 137: HAART term is not explained in full.

In the manuscript, after the term HAART, we quoted the full term, highly active antiretroviral therapy, as the additional term that was searched. This can be found on line 143 and 144.

  • . Line 164: The 5 compounds should not only be mentioned in the text but also their negative values reported.

                  This was a typo. There are only 4 compounds identified here. The compounds and their respective values have been added as text in the manuscript.

  • . Line 170: Why was the CNS-MPO index calculated and reported for only 4? Where is the fifth compound from the graph above?

                  This was a typo. There were only 4 compounds and only 4 CNS-MPO values.

  • Line 181: Explain in more detail the significance of positive and negative values that identify effective and ineffective drugs.

Assuming this is referring to the CNS-MPO score, this score ranges from 0 to 6, so negative values are not possible. A values of 4 or greater has been shown in the literature to correlate with adequate penetrance to the CNS.

  • Line 183: Provide better details on how these drugs were chosen.

The 4 medications that were chosen for the CNS-MPO score were chosen because they were the only ones that demonstrated a negative sRGES value. This is described in the text of the manuscript.

  • . Line 193: Only one drug from those cells remained. Where did the other drugs come from?

Lamivudine was the only medication from the screen that we were able to obtain for testing. Unfortunately, we were not able to obtain the other ones due to logistical reasons.

  • . Line 226: 20M concentrations of drugs? Probably the word "micron" disappeared due to formatting, but how was this dose chosen? And how is it related to what was predicted by the drug repurposing approach?

This is a typo, it has been corrected in the manuscript. The dose was chosen based on similar non-lethal dosing from in vitro experiments using ART.

  • . How is Figure 4a different from Figure 1a?

Figure 1a depicts a heatmap of several antiretroviral medications and their affect on the viability of several established glioma cell lines. Figure 4a depicts a heat map of the synergistic potential of different combinations of antiretroviral compounds. The descriptions of each respective figure have been updated.

  • . Line 254: Figure 4b is useful in a review, but here it seems out of context. Authors should report results and their discussion.

Thank you for your comment. This figure would be useful in a review, however we feel it adds a valuable visual representation in this manuscript. In particular, the effects on stemness are relevant to our findings in this manuscript.

  • . Line 273: In one clinical trial, there is no response rate, so writing that it ranged from 11 to 65% is not correct.

The wording in the manuscript has been changed to indicate there were only 2 that showed a response rate.

Reviewer 4 Report

Comments and Suggestions for Authors

The manuscript addresses the study of pleiotropic anticancer effects of antiretroviral therapy on glioblastoma, for which a comprehensive drug repurposing analysis of in vitro cellular effects and rationalization resorting to bioinformatic tools was carried out.

Despite the interest in the approach, some considerations should considered by the authors, specifically:

The novelty of this study should be better documented in the Introduction part.

The synergistic effect of ART should be documented.

The differences in the ART in the different cell lines should be discussed in more detail. 

Author Response

The novelty of this study should be better documented in the Introduction part.

Thank you for the feedback. The novelty in our study lies on the use of existing data on ART in GBM cell lines to perform an unbiased screen that is validated by our own experiments. We expanded on this in the introduction as suggested.

The synergistic effect of ART should be documented.

Thank you for your comment. We did not assess the synergistic effect of ART in our in vitro models. We briefly discussed the use of multiple drug regimens as standard therapy in patients with HIV, however, did not intend to asses the synergistic effect of multiple drug regimens on glioma cell lines in this study.

The differences in the ART in the different cell lines should be discussed in more detail. 

Thank you for your comment. The two drugs that were used for our in vitro validation were lamivudine and abacavir. There were no differences in ART on these cell lines.

Round 2

Reviewer 4 Report

Comments and Suggestions for Authors

The scientific notation must be corrected.